# Alterations of Plasma Pro-Inflammatory Cytokine Levels in Children with Refractory Epilepsies

**DOI:** 10.3390/children9101506

**Published:** 2022-10-01

**Authors:** Tatia Gakharia, Sophia Bakhtadze, Ming Lim, Nana Khachapuridze, Nana Kapanadze

**Affiliations:** 1Department of Childs Neurology, Tbilisi State Medical University, 0186 Tbilisi, Georgia; 2Evelina London Children’s Hospital @ Guy’s and St Thomas’ NHS Foundation Trust, London SE1 7EH, UK; 3Women’s and Children’s Department, Faculty of Life Sciences and Medicine, Kings College London, London SE1 7EH, UK

**Keywords:** paediatric epilepsy, refractory seizures, cytokines, chemokines, neuroinflammation

## Abstract

Paediatric epilepsy is a multifaceted neurological disorder with various aetiologies. Up to 30% of patients are considered drug-resistant. The background impact of interfering inflammatory and neuronal pathways has been closely linked to paediatric epilepsy. The characteristics of the inflamed state have been described not only in epilepsies, which are considered prototypes of an inflammatory pathophysiology, but also in patients with drug-resistant epilepsy, especially in epileptic encephalopathies. The imbalance of different cytokine levels was confirmed in several epileptic models. Chemokines are new targets for exploring neuroimmune communication in epileptogenesis, which control leukocyte migration and have a possible role in neuromodulation. Additionally, prostaglandin E2 (PGE2) is an important effector molecule for central neural inflammatory responses and may influence drug responsiveness. We measured the serum interictal quantitative levels of chemokines (CCL2, CCL4, CCL11) and PGE2 in correlation with the seizure frequency and severity in controlled and intractable childhood epilepsies. Our refractory seizure group demonstrated significantly increased concentrations of eotaxin (CCL11) compared to the controlled epilepsy group. The higher level of CCL11 was correlated with an increased seizure frequency, while the PGE2 levels were associated with the severity of seizure and epilepsy, supporting the findings that proinflammatory cytokines may contribute to epileptogenesis and possibly have a role in developing seizure resistance.

## 1. Introduction

Paediatric epilepsy is a multifaceted neurological disorder with heterogenous aetiologies that affects 0.6% of children aged 0–17 years [1,2,3]. Approximately one-third of patients fail to achieve adequate seizure control [4], particularly those suffering from developmental and epileptic encephalopathies (DEEs) that result in cognitive, motor, or behavioural impairments due to epileptic activity interfering with normal neurodevelopment [5]. Because of the high incidence of drug-resistant seizures in DEEs, there are several alternative treatment options being considered, including the use of corticosteroids, adrenocorticotropic hormone (ACTH), ketogenic diets, vagus nerve stimulators and surgical treatment, each showing varying degrees of effectiveness [6].

It is understood that epilepsy develops as a result of a complex set of pathophysiologic processes involving genetic, structural and metabolic factors, but the majority of epilepsy cases are idiopathic, making it difficult to determine the exact molecular, chemical and genetic mechanisms involved [7]. There is emerging evidence suggesting that aberrant or uncontrolled neuroinflammatory pathways lie beyond eliminated seizure-initiating triggers in the epileptic zones, which cause further seizure propagation and resistance in the developing brain [8]. Characteristics of this inflamed state have been described not only in epilepsies considered prototypes of inflammatory pathophysiology, namely Rasmussen’s encephalitis, genetic epilepsy with febrile seizures plus (GEFS+), temporal lobe epilepsy, limbic encephalitis and epilepsy after traumatic injury, but also in patients with pharmacoresistant epilepsy and in several experimental rodent models of epilepsy [9]. The multiplex interconnections between epilepsy and inflammation in epilepsy research could be assessed through the upregulated inflammatory molecules such as cytokines and chemokines, which mediate the dichotomous correlation between the central nervous system (CNS) and the immune system [10].

Previous studies have implicated various cytokine imbalances in the epileptic tissue as markers of activated cerebral-resident and recruited immune system cell responses [11,12,13,14,15,16,17]. Chemokines are an increasingly recognised new class of molecules involved in the interaction between CNS-resident cells and peripheral immunity. These control leukocyte migration and have a possible role in neuromodulation [18]. It has been demonstrated that some chemokine–receptor pairs, including CCL2, CCL3, CCL4 and CCL11, are highly expressed in hippocampal tissues and have been associated with epilepsy, based on the findings of experimental models of epilepsy and immunohistochemistry of brain tissue samples from patients undergoing surgical treatment for refractory seizures [19]. The stimulation of these receptors can facilitate the release of excitatory neurotransmitters, including glutamate, causing secondary neurotoxicity and nerve cell death [20].

Moreover, it was observed in the blood–brain barrier (BBB) endothelium that in response to the influence of proinflammatory cytokines, cyclooxygenase 2 (COX-2) expression is induced rapidly, which acts as a rate-limiting enzyme for PGE2 production. In turn, PGE2 influences central neural inflammatory pathways [21]. In several rodent models of drug-resistant epilepsy, blockade of the COX-2–PGE2 pathway and prostaglandin E receptor 2 (EP2) transitory inhibition reduced the degree of BBB dysfunction, decreasing the level of cytokine production and causing reactive gliosis and seizure-promoted functional deficits appear, having a role in the anti-inflammatory cascade and neuroprotection [22]. COX-2 and PGE2 receptors may be proposed as targets for new disease-modifying treatments with the goal of improving and restoring seizure pharmacosensitivity and advancing therapeutic outcomes [23].

However, the reported studies are mostly based on experimental evidence about immediate postictal changes in proinflammatory markers, but the definite burden of their alteration in the genesis and phenotype of epilepsy is unclear. Nonetheless, the research on the role of inflammation in the pathogenesis of Developmental and Epileptic Encephalopathies is of particular clinical importance. The currently available anti-seizure medications are neither preventive nor curative and have multisystem side-effects, including reduced cognitive and memory function, as well as behavioural problems in children [24,25,26,27,28,29,30]. Additionally, the abovementioned proinflammatory markers are one of the plentiful groups that are keenly overexpressed in epileptic zones, implying that via targeted anti-inflammatory immunotherapies these may be novel therapeutic keys in the future in amending mechanisms of epileptogenesis to prevent seizure recurrence and refractoriness in epilepsies [31]. We measured in sera comparably unknown proinflammatory factors—namely chemokines (CCL2, CCL4, CCL11) and PGE2—in children with controlled and refractory epilepsy, aiming to explore the correlation between the interictal cytokine quantitative levels with the seizure frequency and severity and the duration of epilepsy from bench to bedside.

## 2. Materials and Methods

We prospectively enrolled patients aged 0–16 years from the clinical setting of Givi Zhvania Academic Clinic of Paediatrics, Tbilisi, Georgia, during the period between March 2019 and April 2021. Ethical approval was obtained from the research ethics committee of the medical faculty at Tbilisi State Medical University (10 November 2019). The study rationale was justified, and written informed voluntary consent was obtained by parents or legal carers. Information was provided to research subjects stating that the study was to be freely given and that it would not impart a medical benefit to them; they voluntarily confirmed their willingness to process the obtained data.

The collected medical history data covered the patients’ demographics, neurodevelopmental parameters, epilepsy duration, achieved seizure control, treatment plans with anti-seizure medications and previous investigations. The International League Against Epilepsy (ILAE) Classification 2017 was applied for the seizure semiology. The seizure frequency information was counted according to the patients’ seizure diaries. For the evaluation of the seizure severity, the National Hospital Seizure Severity Scale (NHS3)—a refined version of the Chalfont Seizure Severity Scale—was used [32]. In the timeframe of the last three months before a patient visit, we applied questions for clinical judgement by scoring 0–4 points in each component of the NHS3 scale—seizure types, seizure duration, falls, urine incontinence, warning, loss of consciousness and recovery time. For each type of seizure, a total of more than 15 points was considered severe. In addition, for the purpose of defining the epilepsy severity overall, we used the single-itemed Global Assessment of the Severity of Epilepsy (GASE) scale, which is a seven-point Likert scale describing clinical characteristics ranging from “not severe at all” to “extremely severe”, while considering all features of the patient’s epilepsy medical history. The seven aspects are: frequency of seizures, intensity of seizures, falls or injuries during seizures, duration or severity of the post-ictal period, total dose/number of anti-epileptic drugs (AEDs), side effects of AEDs and interference of epilepsy or drugs with daily life activities [33].

The patients with drug-resistant epilepsy were selected for study group 1, controlled epilepsy patients for study group 2 and afebrile non-epileptic controls for study group 3. For the controlled epilepsy group, we only selected children who had been seizure-free for at least one year and had proven epilepsy as assessed via electroencephalographic recording (EEG) and clinical assessment. In the refractory seizures group we enrolled interictal patients having recurrent seizures while being treated with two first-line antiepileptic drugs with proven epileptic encephalopathies as assessed via EEG and clinical assessment, according to the ILAE definition of refractory seizures [4]. Randomly chosen healthy volunteers were selected as the control group. The exclusion criteria for the study groups included the following: fever or fever-causing recent CNS or somatic disease, acute infections, autoimmune disorders and allergies. To minimise the influence of acute ongoing seizures on the cytokine release kinetics, patients with suspected seizures or with non-epileptic paroxysmal disorders were excluded.

All patients underwent systemic and neurological assessments. Next, electroencephalograms (EEGs) were conducted. In the following stage, blood samples were taken, prepared and sent to the lab.

### 2.1. Laboratory Investigations

All tests were done on venous blood samples. To exclude systemic inflammation, a few general systemic tests were performed, including a full blood count (FBC), C-reactive protein (CRP) and blood glucose levels. For the cytokine-level determination after 1 h to leave time for blood to clot at room temperature, serum samples were obtained by centrifugation (15 min, 3000× *g* rpm). Thereafter, the sera were stored at −80 °C and transferred to the laboratory for further assessment for the chemokines CCL2, CCL4 and CCL11 and PGE2 via enzyme-linked immunosorbent assay (ELISA). (Provided kit and analysers: PIA ELISA—HCYTA-60K, Millipore; Luminex, Millipore; KGE004B, R&D—Tecan Sunrise, Tecan.)

### 2.2. Statistical Analysis

The statistical analysis of the cumulated data was conducted using SPSS 21 (IBM SPSS Statistics, version 21.0, Armonk, NY, USA). The continuous variables with a normal distribution are presented as means ± SD standard deviation; non-normal variables are reported as the median interquartile range (IQR). The quantitative data for the cytokine concentrations tested via a Shapiro–Wilk test showed a non-parametric distribution. First, the Kruskal–Wallis test was run and then post hoc the Mann–Whitney U test was used for a comparison of the concentrations within study groups. Spearman’s rank correlation was used to correlate the quantitative cytokine levels with the seizure severity and frequency and epilepsy duration. Here, *p*-values < 0.05 were considered statistically significant.

## 3. Results

The 56 patients participating in the study were arranged into three groups: patients with drug-resistant epilepsy (N = 20), patients with pharmacosensitive controlled epilepsy (N = 20) and a healthy control group (N = 16). From the patients included in the refractory seizure group, two patients were diagnosed with Landau–Kleffner syndrome, two with tuberous sclerosis or Lennox–Gastaut syndrome, one with Rett-like syndrome (STXBP1 epilepsy), two with Dravet syndrome and one with West syndrome. Others were included without the specific epileptic syndromic diagnosis but considered as drug-resistant following their treatment outcomes according to the ILAE criteria for refractory seizures [34].

The average duration of disease in the seizure-controlled group was 3.8 ± 1 years and in the refractory epilepsy group was 6.5 ± 0.5 years. In the drug-resistant seizure group, none of our patients were treated with a ketogenic diet or had a vagus nerve stimulator. Most of them were on triple AEDs. In the context of seizure severity among our patients, the NHS3 score ranges in observed different types of seizures were as follows: 1–3 for myoclonic seizures; 3–4 for typical absence; 15–18 for atonic seizures; 21–25 for bilateral tonic-clonic seizures; 10–15 for behaviour arrest; 5–8 for epileptic spasms. In the context of epilepsy severity, the applied GASE scoring for children within refractory group showed that 20% of them had moderately severe epilepsy at baseline, 35% were quite severe and 50% were extremely severe. The patient data regarding the demographics, clinical characteristics and findings from the neuroradiological and neurophysiological investigations are outlined in Table 1.

To exclude systemic inflammation versus seizures, a few general systemic laboratory tests were performed. No significant statistical differences were found in the interpretation of systemic laboratory findings among the different study groups (all *p* > 0.05). Data for the CRP, glucose, WBC and neutrophil-to-lymphocyte ratio (NLR) in each study group are presented in Table 2.

Our analysis of the targeted cytokine levels showed that CCL11 was higher in the drug-resistant seizure group compared to the controlled seizure group (U = 54.5, *p* = 0.014). The median (IQR) serum CCL11 levels were 118.0 mg/dL in the drug-resistant group and 66.4 pg/dL in the controlled epilepsy group. The median values for CCL2, CCL4 and PGE2 were not significantly different across the study groups (all *p* > 0.05). Figure 1 and Table 3 demonstrates the statistical analysis results for each cytokine.

Next, we correlated the seizure frequency and severity and epilepsy duration with the levels of targeted proinflammatory cytokines. We found that the CCL11 concentration was significantly correlated with the seizure frequency (Rs = 0.786). The PGE2 concentration showed a strong correlation with the seizure severity (Rs = 0.886). The epilepsy duration was not correlated with the levels of any of the explored cytokines (all *p* > 0.05). The correlation analysis results between the seizure severity and frequency, epilepsy duration and cytokine concentration in study groups are shown in Table 4 and Figure 2 and Figure 3.

## 4. Discussion

The potential role of inflammation in the pathogenesis of epilepsy has gained increasing attention during the last decade, and the background impact of interfering inflammatory and neuronal pathways is closely linked to paediatric epilepsy. The characteristics of the inflamed state have been described especially in DEE epilepsies [5]. The DEEs include West syndrome, epilepsy with myoclonic–atonic seizures, Landau–Kleffner syndrome, Dravet syndrome and Lennox–Gastaut syndrome [34]. As drug-resistant epilepsy presents in DEEs, an inflammatory theory arose from the fact that corticosteroids can successfully be used for the treatment of resistant seizures. Regarding the role of neuroinflammation in our understanding of epileptogenesis, it is worth noting some well-established immune privileges of the CNS, including a lack of resident dendritic cells (DCs), modified inflammatory processes and the haematoencephalic barrier, as well as a trend of hematogenous cell recruitment in the CNS [35]. Moreover, it has recently been shown that the immune and nervous systems have to share groups of inflammatory molecules, such as certain chemokines and cytokines [16,18,36]. Walker et al. showed that dynamic interactions between cytokines and chemokines direct the immune responses in the area of inflammatory foci, thereafter interfering with the mechanisms of epileptogenesis [37]. The imbalance of different cytokine levels was confirmed in the epileptic tissues, sera and cerebrospinal fluid [38]. Here, we attempted to explore comparably unknown interictal proinflammatory cytokine serum quantitative levels in correlation with the seizure frequency and severity and epilepsy duration as indicators of refractoriness in the different clinical settings of childhood epilepsies.

Supporting the findings that a group of proinflammatory molecules known as chemokines contribute to epileptogenesis and may have a possible role in seizure repetitiveness and refractoriness is still disputable because the clinical information is scarce. Chemokines were initially identified as being involved in the control of leukocyte traffic, but there is now a growing understanding that chemokines and their receptors are key components in CNS homeostasis, regulating such diverse processes as neurodevelopment, synaptic transmission and neuroinflammation, among others [39]. Since chemokines serve as immune system regulators, they can play an important role in coordinating nerve–immune communication in conditions such as brain injury, inflammation, hypoxia–ischemia and epilepsy. Recent neuroinflammation studies have suggested that specific chemokines and their receptors are responsible for trafficking some blood-derived leukocytes into an inflamed CNS during experimental autoimmune encephalomyelitis (EAE), epilepsy and other model conditions of CNS injury. In subsequent experiments, CCL2 signalled to the CCR2 receptor to recruit monocytes to the inflamed area, and CX3CR1 (C-X3-C motif chemokine receptor 1) was demonstrated to be essential for NK cell recruitment [40]. As was shown, the extravasation of leukocytes is mediated by chemokines, which cluster integrins and change their conformation, leading to high affinity and tight interactions between vascular cell adhesion molecules and integrins [18]. An endothelial cell’s luminal surface, which is immobilised with chemokines, generates a signal that triggers the arrest of leukocytes rolling under flow. Leaked through a disrupted BBB, the peripheral leukocyte interaction with CNS-resident immune cells may have a contributing role in enhancing seizure susceptibility [41,42]. The inhibition of white blood cell adhesion in neurovascular circulation, alternatively via the genetic desensitisation of adhesion receptors or inhibition of the interconnection by neutralising antibodies, repeals the seizure induction and development of epilepsy [43]. An impaired BBB may not be the only result, but this has been considered an involved mechanism of initiating and propagating seizures [44]. A recent study also indicated that the transport of chemokines through the haematoencephalic barrier is facilitated [45], suggesting that peripheral proinflammatory molecules hasten the development and continuation of neuroinflammatory chains through the underlying impaired BBB pathophysiology.

Comparing our two epilepsy groups, we found higher concentrations of CCL11 in the group of drug-resistant patients compared to the controlled epilepsy group. The increase in CCL11 levels was associated with the seizure frequency, while no significant differences were found among the levels of other chemokines.

Among the chemokine family, eotaxin CCL11 acts as an active chemoattractant for eosinophils, and its clinical role has mainly been discussed in allergic diseases, but as reported previously, it also mediates the migration of microglia towards neuroinflammatory sites, where it triggers reactive radical production from microglia by stimulating microglial neurotoxic factors such as the nicotinamide adenine dinucleotide phosphate oxidases (NOX1, NOX2), tumor necrosis factor-α (TNF-a) and interleukin-1b (Il-1b), which leads to the facilitated release of glutamate [46]. It is suggested that the key mechanism promoting CCL11-mediated oxidative stress propagation and glutamate excitotoxic neuronal death is the NOX1 pathway. This mechanism could be a possible explanation for CCL11’s proconvulsant effect, and a variety of neurological disorders have been associated with an increase in microglial NOX1 expression [47].

Consistent with our results, a few studies have shown that in patients with neuroinflammatory diseases, the serum and liquor samples of CCL11 levels were higher, including in neuromyelitis optica spectrum disorders [48]. More interestingly, in the study of the correlation of certain chemokines and cytokines’ expression in the brain, the different anatomical regions showed epilepsy-related eotaxin differences [19]. In particular, the perivascular astrocytes secrete CCL11; next, in response to this, the microglial cells upregulate the expression of the receptors CCR2, CCR3 and CCR5, acting as receptors for CCL11, especially in the hippocampus compared to the entorhinal and temporal cortices. Ragozzino et al. showed that chemokine signalling might play a role during hippocampal development as a consequence of the activation and ramification of microglial cells [49]. The development of glia–neuron communication in these observations supports the hypothesis that chemokines are involved in adjusting hippocampal synaptic plasticity and possibly contributing to the mechanism of the seizure recurrence.

Furthermore, Parajuli et al. showed that CCL11 concentrations in serum and liquor samples showed an upward trend related to age; it is hypothesised that CCL11 may have a suppressing influence on the neurogenesis of the CNS, prompting cognitive impairment via enhanced gliosis, hyperresponsiveness to inflammatory molecules and ROS injury [50,51,52]. Thus, this CCL11-mediated, age-related neuronal damage could represent a potential mirror mechanism of injury or a secondary neuroprotective response in epileptic areas during the chronic initiation of seizures. Unfortunately, we could not establish a correlation between elevated CCL11 levels and the age of patients (Rs(56) = −0.126, *p* = 0.356 Appendix A, Appendix A), nor a link with epilepsy duration.

In addition, we found markedly elevated levels of PGE2 in patients with higher NHS3 seizure scale scores and in patients ranked as having severe epilepsy according to the GASE classification criteria.

It is known that PGE2 is involved in the regulation of membrane excitability, synaptic transmission, plasticity and neuronal survival [53]. The PGE2 levels are known to elevate during and after seizures, the effects of which have been discussed in many studies, while COX-2—the rate-limiting enzyme for PGE2 synthesis—is briskly upregulated for a short period during seizures [54]. In MMA-induced jerks and generalised convulsions, administering PGE2 to animals resulted in a decreased latency of seizure susceptibility and altered EEG amplitude [55]. Preclinical studies have shown that the binding of PGE2 with its receptors, such as EP1-4, located on astrocytes causes astrocytic glutamate exocytosis, secondary hyper excitability, prolonged seizure duration and neuronal cell death in experimental models of neuroinflammatory disorders, amongst them epilepsy [56]. In several rodent models, the EP2 receptor transitory blockade caused reduced seizure-promoted functional deficits and decreased cytokine production, reactive gliosis and the degree of BBB disruption, thereby having marked anti-inflammatory and neuroprotective effects [57].

Moreover, COX-2 and PGE2 receptors may be proposed as targets for new disease-modifying treatments with the goal of improving and restoring the seizure pharmacosensitivity and improving the therapeutic outcomes [58]. Patients responding to valproate demonstrated downregulated activity of COX-2 and correspondingly low plasma levels of PGE2. It is suspected that COX-2 stimulates the multidrug efflux transporter P-glycoprotein (P-gp), meaning that AEDs downregulate COX-2; correspondingly, achieving low plasma levels of PGE2 may suppress the overexpression of P-gp, thereby facilitating drug transport across the BBB and causing improved drug responsiveness [22,59,60,61]. This could be helpful in the interpretation of our results, as we found markedly elevated levels of PGE2 in patients with seizures ranked as severe according to NHS3 scores and lower levels in the controlled pharmacosensitive epilepsy group.

Although the function of PGE2 in epileptogenesis has been studied for a considerable amount of time, there still exist conflicting data concerning it. In the experiments on sarin-induced seizures, analogues of PGE2 were administered immediately after seizures and played a favourable role in reducing brain inflammatory markers; thus, inhibiting PGE2 synthesis with anti-inflammatory medications was not beneficial [62]. While PGE2 and brain-derived neurotrophic factor (BDNF) were highly expressed in the rat hippocampal formation after partial-status epilepticus, this was not the case following generalised-status epilepticus [63]. In the adult dentate gyrus, both of these conditions trigger progenitor cell proliferation at the same level; however, the degree of new neuron survival is lower after generalised-status epilepticus, and the administration of cyclooxygenase (COX) inhibitor has not been shown to influence the survival of new neurons. Thus, the presence of high levels of PGE2 and BDNF is not of importance for the survival of newly formed neurons [63]. It is possible that they regulate the synaptic and cellular plasticity in other ways that interfere with epileptogenesis [64].

Despite previous studies suggesting that CCL2 and CCL4 may have a significant effect on seizure control, which showed elevated levels of chemokines in patients with drug-resistant epilepsy [65], especially in temporal lobe epilepsy [66], as well as recent findings showing that the expression of these chemokines and their receptors in various CNS cells is relevant to psychiatric disorders [67], we were unable to detect alterations of CCL2 and CCL4 levels in our cohort. It was described that CCL2 influences the neuronal physiology by modulating voltage-gated and G-protein-coupled potassium channels, as well as by facilitating the release of certain excitatory neurotransmitters [68]. The activation of CCR4 receptor (CCL4 chemokine receptor 4) expressed in astrocytic and microglial cells increased the release of TNF-α and led to increased glutamate levels [69]. Considering that both of them could have neuromodulatory effects, it should be emphasised that the time course of the cytokine release kinetics likely has a determining role. Considering the remarkable versatility and functional flexibility of chemokines, as well as their proteolytic processing [70], most of the previous evidence has shown that they are elevated in the postictal period shortly after seizures [71]. No significant differences between cytokines were observed among different types of epilepsy in a recent clinical study of 1218 epilepsy patients. It showed that the levels of cytokines are generally not increased in chronic epilepsy patients and that the median concentrations of inflammatory molecules quickly ceased in interictal periods [72]. This is consistent with our interictal findings.

Our study limitations were the small sample size and the assessment of targeted proinflammatory molecules at a single time point. For research questions, we tried to include sufficient controls. We acknowledge that there may have been residual confounders; for example, it cannot be ruled out that the administered AEDs could have a possible specific influence on the levels of cytokines [73]. In the following steps of research, we could reinforce our results with serial measurements from expanded cytokine and chemokine panels and transcriptomics for the same cohort.

## 5. Conclusions

We found that increased CCL11 and PGE2 levels in our patients were correlated with their seizure frequency and epilepsy severity. It has been concluded that our study results support the evidence that suggests a new target system of proinflammatory cytokines for pharmacological intervention that inhibits seizures by interfering with neuroinflammatory pathways. Due to the fact that epilepsy cannot currently be cured, the medical treatments focus primarily on controlling and eliminating seizures. The novel anti-inflammatory drugs in the future might be considered as additional therapy options for suppressing the seizure refractoriness as a means of reducing epileptic disorders, which can result in cognitive, motor and behavioural impairments, affecting children’s quality of life.

## Figures and Tables

**Figure 1 children-09-01506-f001:**
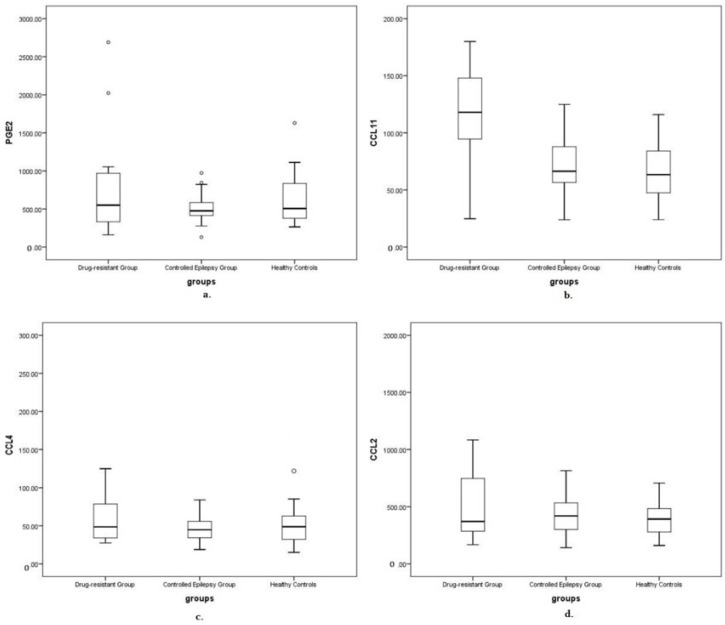
A comparison of serum interictal cytokine levels in drug-resistant, controlled epilepsy and healthy control groups. Kruskal–Wallis test results showing the median, interquartile range (box) and 95% confidence interval of cytokines concentration in patients compared to controls. Measured in pg/mL; ○: outlier. (**a**) PGE2 levels, (**b**) CCL11, (**c**) CCL4 levels, (**d**) CCL2 levels.

**Figure 2 children-09-01506-f002:**
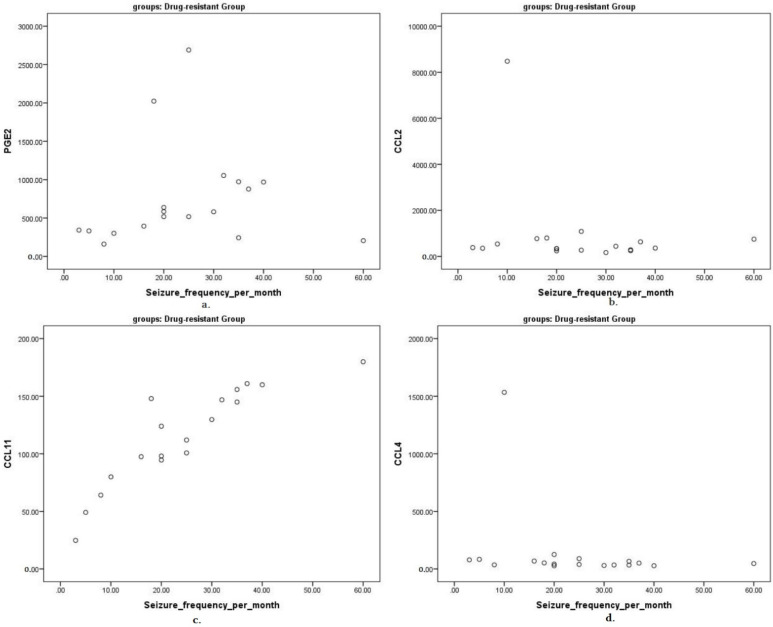
Correlation of targeted proinflammatory cytokines in pg/mL and seizure frequency per month presented as scatterplots: (**a**) PGE2 and seizure frequency; (**b**) CCL2 and seizure frequency; (**c**) CCL11 and seizure frequency; (**d**) CCL4 and seizure frequency.

**Figure 3 children-09-01506-f003:**
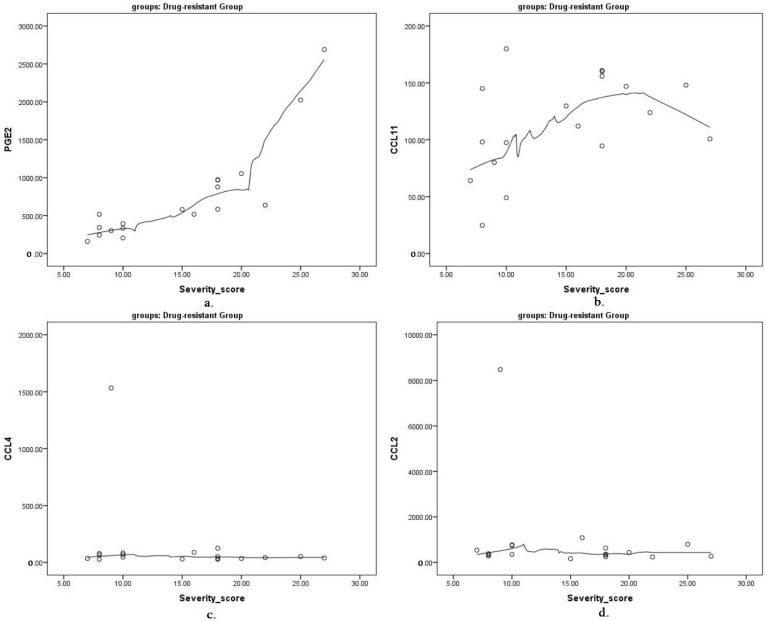
Correlation of targeted proinflammatory cytokines in pg/mL and seizure severity scores presented as scatterplots with Loess fit lines: (**a**) PGE2 and seizure severity scores; (**b**) CCL11 and seizure severity scores; (**c**) CCL4 and seizure severity scores; (**d**) CCL2 and seizure severity scores.

**Table 1 children-09-01506-t001:** Descriptive statistics for the clinical, radiological and electrophysiological characteristics of the study participants. Presented as quantitative data within groups, as percentages and means ± SD.

Study Group	Drug-Resistant Group	Controlled Epilepsy Group	Healthy Controls
N	20	20	16
Female/male	8/12	11/9	8/8
Age (Mean + SD years)	9.2 + 4.2	8.5 + 4.6	9.2 + 4.0
Epilepsy duration n ± SD	6.2 + 0.5	3.8 + 1.2	-
Seizure frequency n ± SD, seizure/month	10.0 ± 14.2	0	-
Brain MRI findings (%)			
Normal	45	68	-
Abnormal	55	32	-
EEG parameters (%)			
Focal interictal discharges	29	26	-
Encephalopathic patterns	46	28	-
Bilateral spike-and-wave discharges	25	46	-
Anti-epileptic drugs (n)			
Mono-drug	2	17	-
Two-drug combinations	10	3	-
Poly-drug	8	0	
NHS score			-
>10	15	3	
<10	5	17	

**Table 2 children-09-01506-t002:** Systemic laboratory findings. Presented as means ± SD.

Study Groups	Drug-Resistant Seizures	Controlled Epilepsy	Healthy Controls
CRP (mg/l)	3.9 ± 4.1	3.6 ± 4.8	3.4 ± 3.3
WBC (10^9^/L)	8.3 ± 4.5	9.4 ± 3.0	7.8 ± 4.2
Neutrophils (%)	57.7 ± 21.6	54.1 ± 22.8	44.1 ± 32.3
Lymphocytes (%)	40.0 ± 16.3	39.6 ± 15.6	41.6 ± 14.7
NLR	1.4 ± 1.3	1.38 ± 1.19	1.03 ± 0.08
GLU (mmol/L)	4.6 ± 2.1	5.3 ± 1.5	4.3± 2.3

GLU—glucose; NLR—neutrophil/lymphocyte ratio; CRP—C-reactive protein; WBC–white blood cells.

**Table 3 children-09-01506-t003:** The interquartile range (IQR) of targeted proinflammatory cytokine concentrations.

Proinflammatory Cytokines	Interquartile Range Quartiles	Drug-Resistant Epilepsy Group	Controlled Epilepsy Group	Healthy Controls
CCL2	Q1	282.0	293.0	168.0
	Q2	**371.0**	**420.0**	**392.0**
	Q3	754.0	561.0	402.0
CCL 4	Q1	33.9	34.0	20.7
	Q2	**48.6**	**44.8**	**38.8**
	Q3	79.7	57.4	44.8
CCL11 *	Q1	90.9	55.9	35.9
	Q2	**118.0**	**66.4**	**53.4**
	Q3	149.9	88.0	74.3
PGE2	Q1	324.2	388.0	278.0
	Q2	**549.5**	**475.0**	**406.0**
	Q3	970.0	595.0	541.0

* Cytokines measured in pg/mL. Q1—25th percentile quartile; Q2—50th percentile quartile; Q3—75th percentile quartile; bolded—median interquartile range; *—indicates significant difference at *p* < 0.05.

**Table 4 children-09-01506-t004:** Correlations between targeted proinflammatory molecules with the seizure frequency per month and with the severity of seizures in drug-resistant epilepsy patients.

Targeted Proinflammatory Molecules	Spearman’s Correlation Rho					
	Seizure Frequency	Sig. (2 Tailed)	Seizure Severity NHS3	Sig. (2 Tailed)	Epilepsy Duration	Sig. (2 Tailed)
CCL2	0.08	0.556	−0.264	0,343	−0.065	0.727
CCL4	0.59	0.663	0.296	0.027	−0.363	0.061
CCL11	0.786 *	0.000	0.185	0.172	−0.106	0.570
PGE2	0.122	0.370	0.886*	0,000	−0.229	0.214

* Significant strong correlation: *p* < 0.05.

## Data Availability

The data presented in this study are available on request from the corresponding author. The data are not publicly available due to study academic institution’s research committee regulations.

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
