# Peer review of "Alterations of Plasma Pro-Inflammatory Cytokine Levels in Children with Refractory Epilepsies"

_children, 2022, doi:10.3390/children9101506_

Round 1
Reviewer 1 Report
1, In the table 2, what’s column of “Title 1, Title 2, Title 3, Title 4”?
2, In the table 3, what’s the meaning of “*” for CCL2?
3, “Figure 3” was mentioned in line 198, but where is it?
4, The “relationship between CCL11 and age” was mentioned in line 290, but where’s the data and result?
5, Each legend should contain the statistical method.
6, For figure 1, which statistical method was used? Mann–Whitney U test is inappropriate for 3 groups.
Author Response
We appreciate the time and effort that you dedicate to providing your valuable feedback on the manuscript. We are grateful for your detailed comments.
We have been able to incorporate changes to reflect most of the provided suggestions. Here is a point-by-point response to your comments and concerns.
- Unfortunately, in table 2, there were extra empty columns of “Title 1, Title 2, Title 3, Title 4” , which were automatically generated from the provided word template draft. Thus, it should be just a table 4x7 Laboratory Systemic Findings (C reactive protein; White Blood Cells, Neutrophils, Lymphocytes, Neutrophil-Lymphocytes Ratio; Glucose) presented as mean ± SD standard deviation for study groups ( drug-resistant, controlled epilepsy, and healthy control groups ).
- In table 3, the symbol “*” was misspointed on CCL2. It should be pointed on CCL11 and indicates a finding of significant difference p<0.05.
- Figure 3 is supposed to represent the graphical view of the data from Table 3-4 for presenting the relationship between seizure severity scores and cytokine concentrations, which is missing from the draft and was overlapping with the figure 2 legend. We are happy to have the opportunity to include it again. Figure 3 is the correlation of targeted proinflammatory cytokines in pg/mL and seizure severity scores presented as scatterplots with Loess fit lines (a) PGE2 and Seizure Severity scores (b) CCL11 and Seizure Severity Score (c) CCL4 and Seizure Severity Scores (d) CCL2 and Seizure Severity Scores
- Correlation between age and CCL11 levels was run post factum to our results and interest from corresponding findings in recent studies, and it was not initially our research question, that’s why it is not included in the result section. The statement is shortly updated with statistical analysis results and data (correlation between age and CCL11 levels, Spearman Rs(56)= -.126, p= 0. 356 ), data represented as a heatmap could be provided in supplements or could be considered as a part of major body text if you would consider the importance of that for the manuscript. (please, see attachement file).*
- For each legend, we updated statistical test descriptions.
- To explore differences between groups, first, we used Kruskal Wallis test, and then, as a post hoc test for straightening differences between two groups, was used Mann Whitney Test. Corresponding findings for three groups according to Kruskal Wallis test are represented in the figure 1 box plot. Apologies for the confusion, thanks to your valuable comment, now we have mentioned about applied Kurskal-Wallis test in methodology although.
We are sincerely sorry for your time regarding the first few comments due to our technical issues in Word editing. I hope that the changes I've made resolve your concerns about the article. I'm more than happy to make further changes to improve the paper and/or facilitate successful publication.
Thank you once again for your time and interest.

Reviewer 2 Report
The manuscript offers very interesting results, but it has important limitations (very well revealed by the authors), so the results may be taken with caution, but anyway in my opinion they may be published.
Author Response
Thank you for reviewing our manuscript.
We are grateful for your valuable comments and for considering our work for publication. Within the frames of the fundamental research grant, we are pioneering this type of study from the field in our cohort. We thank you for honestly evaluating and considering our point of view in limited circumstances.
We are more than happy to make further changes to improve the paper and/or facilitate successful publication.
Thank you once again for your time and interest.
Round 2
Reviewer 1 Report
All my concerns have been addressed by the authors. I would like to recommend the paper for publication.